# Variation in prevalence of participation limitation, injury location, and training habits between young and master runners: A cross-sectional study

Rachel Berns[1]*, Jo Armour Smith[2], Natalia Sánchez[2,3], Samantha Jeffcoat[2], Susan Sigward[1], Andrew Hooyman[2], Shawn Farrokhi[2]

1 Division of Biokinesiology and Physical Therapy, University of Southern California, Los Angeles, California, United States of America, 2 Department of Physical Therapy, Crean College of Health and Behavioral Sciences, Chapman University, Irvine, California, United States of America, 3 Department of Electrical Engineering and Computer Science, Fowler School of Engineering, Chapman University, Orange, California, United States of America

* rberns@usc.edu

## Abstract

### Background/Objectives

Running is a popular form of physical activity, but its growing participation has been accompanied by high rates of injury that may limit ongoing engagement, particularly in older runners. This study explored differences in running participation limitation, injury location, and training characteristics between young (18–34 years) and master (>35 years) runners.

### Methods

A total of 207 runners (105 young, 102 master) completed an online survey assessing demographics, training habits, and injury-related participation limitations using the updated Oslo Sports Trauma Research Center Overuse Injury Questionnaire.

### Results

Master runners reported higher rates of injury-related participation limitation than young runners (51% vs. 34%, p = 0.04) and more upper extremity injuries (p < 0.01). They also had longer running histories, higher weekly mileage, and longer training durations (p < 0.05). Among master runners, increased odds of running participation limitation were associated with smoking [Odds Ratio (OR)=1.89], running on gravel/ pebbled surfaces (OR=1.10), and owning more pairs of running shoes (OR=1.02), with no protective factors identified. Among young runners, reduced odds of running participation limitation were associated with running on off-road tracks (OR=0.68), having a running coach (OR=0.70), wearing shoes with special features (OR=0.75),

**Data availability statement:** All data used for this manuscript can be accessed at https://osf.io/ev8sc/overview.

**Funding:** The author(s) received no specific funding for this work.

**Competing interests:** The authors have declared that no competing interests exist.

and training on multiple surfaces (OR=0.79), while slightly increased odds were associated with longer average training sessions (OR=1.04) and running on grass (OR=1.08).

## Conclusion

These findings highlight the high prevalence of running participation limitations, particularly in master runners, and support the need for age-specific training and injury prevention strategies.

## Implications

Different age groups of runners may benefit from tailored risk mitigation strategies: young runners may benefit from coach-guided training, appropriate footwear selection, and varied training surfaces, while avoiding unnecessarily long sessions. For master runners, emphasis may be placed on screening for modifiable health risks (e.g., smoking), careful management of training load and surface exposure, and addressing upper-extremity complaints, given their higher prevalence of participation limitation.

## Introduction

Running is one of the most widespread forms of physical activity globally, attracting individuals across a spectrum of participation, from casual recreational runners to competitive athletes [1]. Since 2020, the U.S. has seen a rise in recreational runners, along with increased running volume and frequency among experienced runners, partly due to the COVID-19 pandemic and the demand for exercise options that aligned with public health guidelines [2]. Much of the appeal to take up running as a form of exercise lies in its accessibility. Running is low entry-level and requires minimal financial investment, making it a suitable activity for highly heterogeneous groups of people [3]. Given its broad appeal and health benefits, understanding the factors that influence running participation and associated injury risks is increasingly important.

The recent surge in running participation parallels an increase in running-related injuries (RRIs) [2]. Research has indicated that approximately 50% of runners will experience a participation-limiting RRI each year, highlighting the need for preventive measures [4]. A recent systematic review reported that the overall prevalence of RRIs among runners is as high as 45%, with the knee, ankle, and lower leg diagnoses accounting for the highest proportion of RRI prevalence [5]. To this end, multiple factors have been identified to influence the rates of RRIs in runners. Demographic factors such as age, gender, and body weight have been shown to be associated with the presence of RRIs [6–9]. Training variables such as stretching before and after running [10], warm-up and cool-down routines, and running on hard surfaces can also influence both injury occurrence and severity [11–12]. Other factors, such as

adherence to proper shoe usage, including the replacement of running shoes based on mileage rather than wear, can also influence the risk of RRIs [13]. Together, these findings underscore the multifactorial nature of RRIs and the importance of investigating how modifiable and non-modifiable factors interact to influence injury risk and participation limitations. However, despite extensive literature describing RRI incidence and associated risk factors, comparatively few studies have directly examined to what extent running participation is limited by run-related injuries, and how these limitations differ by age, particularly in direct comparisons of young versus master runners.

Master runners, defined as runners aged 35 and older [14–15], are a distinct subset of the running population that are thought to engage in running for unique reasons beyond fitness alone. This age cutoff is consistent with common competitive classifications for "masters" athletics events held by organizations such as the World Masters Athletics [16] and the USA Track & Field [17]. Previous research has corroborated that physical health benefits underscore running motivations for runners of all ages; however, master runners often place further emphasis on the social connections and emotional fulfillment derived from running [18]. These personal motivators may enhance the likelihood of sustained running participation over time, yet they are at odds with the higher injury rates experienced by older runners. Previous research has shown that master runners experience significantly higher overall injury rates compared to younger runners, with a greater prevalence of multiple areas of pain and musculotendinous injuries, particularly in the plantarflexor (calf) muscles, Achilles tendon, and hamstrings [19]. Age-related changes in aerobic capacity, muscle strength/elasticity, as well as running biomechanics, may also influence how loads are distributed during running and, in turn, contribute to differences in injury risk and participation limitation across age groups [14,20–25]. These age-related differences in injury location and severity, along with differences in running habits such as higher weekly mileage, more frequent training sessions, and greater orthotic use suggest that master runners may face distinct risks compared to their younger counterparts [19]. This highlights the need to examine how age influences running-related injury patterns, training behaviors, and participation limitations by directly comparing young and master runners.

The objectives of the current study were 1) to examine the differences in rates of running participation limitations and injury location between young and master runners, 2) to evaluate whether demographics, social factors, training characteristics and running habits are different between young and master runners, and 3) to determine what factors influence the risk of running participation limitation in young and master runners. We hypothesized that running participation limitation rates would be higher for master runners and that the location of their RRIs, as well as their run training characteristics and habits, would be different from those of younger runners. In addition, we hypothesized that the factors associated with reduced or increased risk of running participation limitation would be different between young and master runners.

## Methods

### Participants

This study was conducted via an online survey completed by 207 adult runners from various regions across the United States from May 8th, 2024, to June 19th, 2024. Participants responded to recruitment flyers distributed through social media platforms, university campuses, emails to running clubs, and word-of-mouth referrals. To be eligible, individuals had to be at least 18 years old and currently engaged in a regular running program (defined as running 1–3 times per week for at least the three months prior to the survey), regardless of their level of experience. Given the broad inclusion criteria, all individuals who responded met eligibility requirements and completed the full survey; no participants were excluded or left the survey incomplete. Based on age, participants were categorized into two nearly equal groups: 105 young runners (aged 18–34 years) and 102 master runners (aged 35 years and older). All participants provided written informed consent before completing any study surveys, and the study received approval from the Chapman University Institutional Review Board (IRB-24–138).

## Questionnaires and surveys

A previously developed survey [26] focusing on demographics and running characteristics and habits in Brazilian runners was converted into an electronic format and distributed via the protected web application REDCap (Research Electronic Data Capture) to all participants. To maintain the integrity of the original instrument, we retained the same core content domains (participant characteristics, running history/training characteristics, and equipment habits) while making minor adaptations to improve relevance to our U.S.-based recreational running sample and to align the survey with our study outcomes. The survey comprised three sections: (a) questions related to participants' demographics and social behaviors, including age, sex, race, weight, height, education level, participation in other sports, and history of smoking or vaping; (b) questions regarding participants' running history and characteristics, such as the number of years they have been running, frequency of weekly running sessions, duration of training runs, mileage per training run, average weekly running mileage, preferred running races, use of special insoles, and number of running shoes in use; and (c) running training habits, including frequency of stretching, use of warm-up and cool-down routines, and the frequency of running sessions on various surfaces. Adaptations that were made to the survey [26] included converting units and response options for clarity (e.g., miles/inches/pounds; expanded race-distance options), adding select items relevant to this population (e.g., vaping history, orthotic use, and additional demographic/health questions), and modifying how injury was captured. Because the demographics/training-habits items are descriptive rather than a single latent construct, internal consistency statistics (e.g., Cronbach's alpha) were not calculated.

To assess limitations in running participation due to overuse injury and pain, we utilized the updated Oslo Sports Trauma Research Center Overuse Injury Questionnaire (OSTRC-O2) [27]. This replaced the injury definition and text-based injury-history questions used in the original survey tool [26], allowing participation limitation to be quantified over the prior seven days using a validated overuse injury tool. The OSTRC-O2 quantifies injury severity and its impact on participation in training or competition, modifications to training, performance limitations, and pain experienced during athletic activities over the last seven days. The OSTRC-O2 score is derived from four key questions, with a score of zero indicating no pain and full participation, while a score of 100 indicates severe pain and inability to participate. Because OSTRC-O2 severity scores typically show a strong floor effect (many 0 scores) and a zero-inflated, right-skewed distribution in athlete monitoring studies [28–29], and because the total score is derived from non-linear/uneven response-category weighting, we a priori dichotomized the outcome (0 vs > 0) to distinguish runners with no participation limitations from those reporting any injury-related problem/participation limitation over the last seven days.

The location of overuse injuries in runners was determined using a standardized body pain diagram, enabling participants to visually identify the specific area(s) of their pain. Participants completed an electronically delivered body pain map with predetermined body locations and directly selected the location(s) corresponding to their symptoms, rather than providing text-only location descriptions, to improve clarity and reduce ambiguity in symptom location reporting. The body pain map captured symptom location only; pain intensity and pain quality (e.g., sharp, aching, burning, numbness/tingling) were not recorded. An RRI was defined as current musculoskeletal pain that restricted or halted running (in terms of distance, speed, duration, or training) for at least seven days, affected three consecutive scheduled training sessions, or necessitated consultation with a physician or other health professional [30]. Participants could select more than one body location/region if they experienced pain in multiple areas (i.e., multiple RRIs or multi-site symptoms). Each runner was instructed to identify the body area(s) corresponding to their RRI(s). Because injury location was directly selected by participants from predefined locations (rather than interpreted and coded by researchers from freehand markings), inter- and intra-rater reliability testing was not applicable. This visual approach is favored by both patients and clinicians, as it facilitates a more accurate and objective assessment of symptom distribution [31–32]. The marked areas were subsequently categorized from 48 possible distinct body locations into eight mutually exclusive regions: 1) foot and ankle; 2) shin; 3) calf; 4) knee; 5) hip and thigh; 6) pelvis and low back; 7) upper back, ribs, and neck; and 8) shoulder, elbow, and hand.

## Statistical analysis

All categorical variables were individually cross-tabulated with the two age categories (young/master) using chi-square tests of independence. Differences between groups for all continuous variables were compared using independent sample t-tests or Wilcoxon rank sum tests. To compare the rate of running participation limitation between young and master runners, we categorized participants based on their responses to the OSTRC-O2 questionnaire. Those reporting no RRIs, indicated by a score of zero on the OSTRC-O2, were classified as having "No participation restriction" over the previous seven days. In contrast, participants with any non-zero score, indicating reduced participation to complete inability to participate in running during the previous seven days, were classified as having "Minor to severe participation restriction." Group differences in participation restriction between young and master runners were analyzed using a chi-square test.

To identify and evaluate the variables with the most robust association between demographic characteristics, social factors, and training habits with participation restriction status, we performed a generalized LASSO (Least Absolute Shrinkage and Selection Operator) regression [33–34] using the previously defined binary response categories ("No participation restriction" vs. "Minor to severe participation restriction") based on OSTRC-O2 scores. Two separate LASSO logistic regression models were constructed, one for young runners and one for master runners, to assess these associations within each cohort. The LASSO method applies an L1 penalty to the regression coefficients, promoting sparsity by shrinking less important variables to zero, thereby identifying a minimal set of influential predictors. This approach is well-suited for handling correlated and high-dimensional predictors. Model tuning was conducted using stratified 10-fold cross-validation to select the optimal value of the regularization parameter ($\lambda$), that achieved the highest average area under the curve across all folds. All continuous predictors were standardized (mean = 0, standard deviation = 1) prior to modeling. By design, LASSO mitigates concerns of multicollinearity, allowing for robust variable selection even in the presence of interrelated predictors. Accordingly, some variables that differ between groups in univariate analyses may not be selected if they do not provide unique incremental predictive value beyond correlated predictors under the cross-validated penalty. Additionally, because models were fit separately within each cohort, between-group (young vs. master) univariate differences are not necessarily informative of which predictors are retained for within-cohort participation-restriction prediction. These results reflect the variables selected by the LASSO regression model, which prioritizes parsimony and prediction over statistical significance testing. Because LASSO regression applies an L1 penalty that alters the distribution of coefficient estimates, traditional p-values and 95% confidence intervals are not directly available. Despite this limitation, LASSO remains preferable to stepwise regression because it provides more stable variable selection, handles multicollinearity among predictors more effectively, and reduces overfitting through cross-validated penalization rather than relying on the unstable, data-driven inclusion and exclusion decisions characteristic of stepwise methods. Therefore, the interpretation of results focused on the strength and direction of associations based on odds ratios (OR) rather than statistical significance [34]. We also conducted sensitivity analyses to assess the stability of the cross-validated $\lambda$ values for both age-group models by varying each optimal $\lambda$ by ±20% and examining whether the same coefficients were retained across these perturbations. All analyses were performed using R version 4.4.0.

## Results

### Differences in participation limitation between young and master runners

A greater proportion of master runners (51%, 52 out of 102) reported OSTRC-O2 scores above zero compared to younger runners (34%, 36 out of 105), indicating some level of reduced or impaired running participation over the previous seven days (p = 0.04).

### Differences in location of pain between young and master runners

Overall, the knee (27%−28%) and the foot/ankle (26%−27%) were the most frequently reported pain regions for both young and master cohorts, with no between-group differences (Fig 1). Of note, only a greater proportion of shoulder/arm/forearm injuries in master runners was significantly different from younger runners (13% vs. 2%; p = 0.008). Proportions across other body regions were not significantly different between age groups (Fig 1).

### Differences in demographics and social habits between young and master runners

Table 1 presents the demographic and social characteristics of young and master runners. As expected, master runners were significantly older than younger runners (mean age: 43.7 vs. 25.5 years, p < 0.001). Racial composition also differed significantly between the two cohorts (p = 0.002), with a higher proportion of master runners identifying as White (58% vs. 48%) and Black or African American (28% vs. 14%), while a higher proportion of young runners identified as Asian (18% vs. 9%), or more than one race (8% vs. 2%). Educational attainment also differed significantly between groups (p < 0.001), with younger runners more likely to have a high school diploma, whereas master runners more commonly held master's degrees or associate degrees. Although traditional smoking was more prevalent among master runners (12% vs. 4%), this difference did not reach statistical significance (p = 0.06). However, e-cigarette use was significantly more common among young runners (36% vs. 22%, p = 0.03). There was no significant difference between groups in terms of sex, body weight, height, or participation in other sports outside of running (p > 0.05).

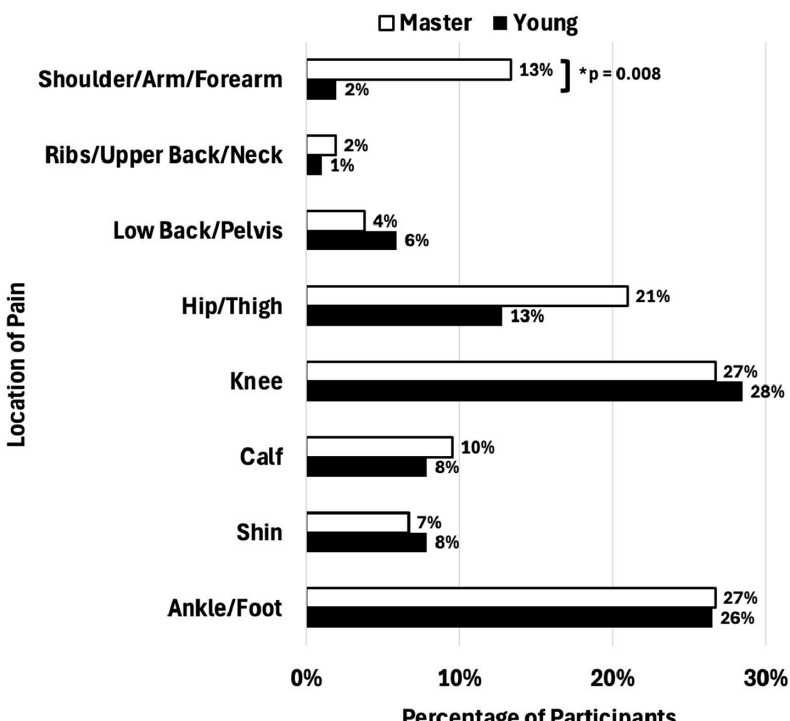

**Fig 1. Bar graph representing the percentage of master and young runners reporting pain in specific body regions.** Master runners are represented by white bars, and young runners are represented by black bars. Because runners could report pain in multiple body regions, percentages in the figure are not mutually exclusive and may sum to more than 100%.

**Table 1. Differences in demographics and social habits between Young and Master runners.**

| Characteristic | Young, n = 105[1] | Master, n = 102[1] | p-value |
|---|---|---|---|
| **Age (years)** | 25.5 (4.2) | 43.7 (9.6) | **< 0.001[2]** |
| **Sex** | | | 0.20[3] |
| Male | 47 (45%) | 56 (55%) | |
| Female | 58 (55%) | 46 (45%) | |
| **Body Weight (pounds)** | 149.4 (26.8) | 155.8 (28.9) | 0.11[2] |
| **Height (inches)** | 67.0 (3.4) | 66.9 (4.2) | 0.80[2] |
| **Race** | | | **0.002[3]** |
| White | 50 (48%) | 59 (58%) | |
| Black or African American | 15 (14%) | 29 (28%) | |
| Asian | 19 (18%) | 9 (9%) | |
| More than one race | 8 (8%) | 2 (2%) | |
| Other | 12 (11%) | 3 (3%) | |
| **Ethnicity** | | | 0.07[3] |
| Hispanic/Latino | 24 (23%) | 11 (11%) | |
| Non-Hispanic | 80 (76%) | 90 (88%) | |
| **Education Level** | | | **< 0.001[3]** |
| High School/GED | 23 (22%) | 2 (2.0%) | |
| Community College/Associate Degree | 2 (1.9%) | 8 (7.8%) | |
| B.S. | 47 (45%) | 48 (47%) | |
| Masters | 13 (12%) | 28 (27%) | |
| PhD/MD | 20 (19%) | 16 (16%) | |
| **Smoking Status (Yes)** | 4 (3.8%) | 12 (12%) | 0.06[3] |
| **E-cigarette (Yes)** | 38 (36%) | 22 (22%) | **0.03[3]** |
| **Participated in Other Sports (Yes)** | 54 (51%) | 41 (40%) | 0.14[3] |

[1]Mean (Standard Deviation); n (%).

[2]Two-sample t-test.

[3]Chi-squared test.

## Differences in running history and training characteristics between young and master runners

Master runners had been running for more years (15.5 vs. 8.1; p < 0.001), had longer average run training sessions (60.5 vs. 50.4 minutes; p = 0.04), and ran a longer average distance per week (31.5 vs. 16.0 miles; p = 0.04) compared to younger runners (Table 2). Master runners also owned more running shoes (3.9 vs. 2.8; p < 0.001) and replaced their shoes more frequently within 6 months (p = 0.04) than younger runners. Additionally, a higher proportion of master runners replaced their running shoes due to mileage versus wear and tear compared to younger runners (p = 0.03), and a higher proportion of master runners had cushions or insoles in their running shoes compared to young runners (51% vs. 32%; p = 0.03). In comparing the type of races each group participated in, a higher proportion of master runners ran in 10 km (35% vs. 20%; p = 0.02) and 10 km to half-marathon (21% vs. 8%; p = 0.01) races compared to younger runners.

## Differences in training habits between young and master runners

Table 3 compares training habits between young and master runners. Overall, most training behaviors were similar between groups, with no significant differences in the frequency of stretching before or after training, warming up, cooling down, or training on common surfaces such as asphalt, off-road tracks, grass, cement, or treadmills (p > 0.05). However, significant group differences emerged for two training surface types. Young runners were more likely to train on gravel or

**Table 2. Differences in running history and training characteristics between Young and Master runners.**

| Characteristic | Young, n = 105[1] | Master, n = 102[1] | p-value |
|---|---|---|---|
| **Years running** | 8.1 (4.9) | 15.5 (15.3) | **< 0.001[2]** |
| **Training sessions per week** | 3.5 (1.4) | 3.6 (2.9) | 0.70[2] |
| **Average training session length (minutes)** | 50.4 (33.6) | 60.5 (38.3) | **0.04[2]** |
| **Average distance per week (miles)** | 16.0 (14.8) | 31.5 (70.9) | **0.04[2]** |
| **Average mile time** | 9.4 (9.6) | 9.8 (7.0) | 0.70[2] |
| **Running coach? (Yes)** | 42 (40%) | 48 (47%) | 0.40[3] |
| **Pairs of running Shoes** | 2.8 (1.6) | 3.9 (2.6) | **< 0.001[4]** |
| **Shoe replacement frequency** | | | **0.04[3]** |
| < 6 Months | 11 (10%) | 23 (23%) | |
| 6 months – 1 Year | 44 (42%) | 45 (44%) | |
| 1 Year – 1.5 Years | 28 (27%) | 23 (23%) | |
| 1.5 years – 2 Years | 19 (18%) | 7 (6.9%) | |
| I don't know | 3 (3%) | 4 (4%) | |
| **Reason to replace shoes** | | | **0.03[3]** |
| Wear and Tear | 86 (82%) | 69 (68%) | |
| Mileage | 19 (18%) | 33 (32%) | |
| **Shoes with special features? (Yes)** | 61 (58%) | 61 (60%) | 0.91[3] |
| **Shoe cushion or insole? Yes** | 24 (32%) | 33 (51%) | **0.03[3]** |
| **Do you wear orthotics? Yes** | 15 (14%) | 23 (23%) | 0.20[3] |
| **What kind of race do you usually run?** | | | |
| Less than 10 kilometers | 54 (51%) | 47 (46%) | 0.50[3] |
| 10 kilometers | 21 (20%) | 36 (35%) | **0.02[3]** |
| >10 kilometers – <½ marathon | 8 (8%) | 21 (21%) | **0.01[3]** |
| ½ Marathon | 28 (27%) | 37 (36%) | 0.20[3] |
| Marathon | 16 (15%) | 17 (17%) | 0.93[3] |
| Ultra marathons | 3 (3%) | 8 (8%) | 0.20[3] |
| Don't run races | 15 (14%) | 6 (6%) | 0.08[3] |

[1]Mean (Standard Deviation); n (%).

[2]Two-sample t-test.

[3]Chi-squared test.

[4]Wilcoxon rank sum test.

pebbles compared to master runners (67% vs. 49%, p = 0.02), while master runners reported training on "other surfaces" more frequently than younger runners (92% vs. 72%, p = 0.01).

## Correlates of running participation limitation in young and master runners

For the LASSO logistic regression analyses, the optimal penalty parameter was identified as $\lambda = 0.089$ for the younger runners' model and $\lambda = 0.119$ for the master runners' model. Among young runners, several variables were associated with reduced odds of participation limitation. These included running on off-road tracks (OR: 0.68), having a running coach (OR: 0.70), wearing shoes with special features such as cushioning, stability, or motion control (OR: 0.75), and training on multiple surfaces (OR: 0.79). Slightly increased odds of participation limitation were associated with longer average training sessions (OR: 1.04) and running on grass (OR: 1.08), though these effects were modest. Among master runners, smoking (OR: 1.89), running on gravel or pebbled surfaces (OR: 1.10), and owning a greater number of pairs of running

**Table 3. Differences in training habits between Young and Master runners.**

| Characteristic | Young, n = 105[1] | Master, n = 102[1] | p-value |
|---|---|---|---|
| **Do you stretch before training?** | | | 0.11[2] |
| Never | 64 (61%) | 50 (49%) | |
| Sometimes/Always | 41 (39%) | 52 (51%) | |
| **Do you stretch after training?** | | | 0.30[2] |
| Never | 78 (74%) | 68 (67%) | |
| Sometimes/Always | 27 (26%) | 34 (33%) | |
| **Do you warm up before training?** | | | 0.91[2] |
| Never | 49 (47%) | 48 (47%) | |
| Sometimes/Always | 56 (53%) | 54 (53%) | |
| **Do you cool down after training?** | | | 0.60[2] |
| Never | 66 (63%) | 59 (58%) | |
| Sometimes/Always | 39 (37%) | 43 (42%) | |
| **Do you train on asphalt?** | | | 0.50[2] |
| Never/Rarely | 83 (79%) | 85 (83%) | |
| Sometimes/Often/Always | 22 (21%) | 17 (17%) | |
| **Do you train on off-road track?** | | | 0.50[2] |
| Never/Rarely | 65 (62%) | 69 (68%) | |
| Sometimes/Often/Always | 40 (38%) | 33 (32%) | |
| **Do you train on grass?** | | | 0.11[2] |
| Never/Rarely | 42 (40%) | 53 (52%) | |
| Sometimes/Often/Always | 63 (60%) | 49 (48%) | |
| **Do you train on gravel/pebbles?** | | | **0.02[2]** |
| Never/Rarely | 35 (33%) | 52 (51%) | |
| Sometimes/Often/Always | 70 (67%) | 50 (49%) | |
| **Do you train on cement?** | | | 0.08[2] |
| Never/Rarely | 75 (71%) | 60 (59%) | |
| Sometimes/Often/Always | 30 (29%) | 42 (41%) | |
| **Do you train on a treadmill?** | | | 0.91[2] |
| Never/Rarely | 63 (60%) | 63 (62%) | |
| Sometimes/Often/Always | 42 (40%) | 39 (38%) | |
| **Do you train on other surfaces?** | | | **0.01[2]** |
| Never/Rarely | 21 (28%) | 5 (8.5%) | |
| Sometimes/Often/Always | 55 (72%) | 54 (92%) | |

[1]n (%).

[2]Chi-squared test.

[3]Wilcoxon rank sum test.

shoes (OR: 1.02) were associated with increased odds of participation limitation. No protective variables were identified for master runners. Sensitivity analyses in which the optimal λ values were varied by ±20% demonstrated stability in the set of coefficients retained by the LASSO models for both younger and master runners. Because LASSO is designed primarily as a prediction-focused variable selection method rather than an inferential statistical tool, the retained coefficients should not be interpreted as evidence of causal relationships, and inferential conclusions beyond the direction and relative magnitude of associations should be made with caution.

 

## Discussion

Running-related injuries are common and can reduce training consistency and overall physical activity participation; however, age-specific evidence comparing participation limitations, injury location, and training behaviors between young and master runners remains limited. Addressing this gap is clinically relevant because age-informed insights may help clinicians and coaches tailor injury prevention, load management, and return-to-running recommendations across the lifespan. Therefore, this study sought to explore how age influences running participation limitations, injury location, and training characteristics by comparing young and master runners. Of note, although the ≥ 35-year threshold is widely used to define "master" runners in both research and competitive classifications, we acknowledge that this cutoff may be physiologically young; thus, the patterns observed here should be interpreted cautiously and should be examined in future studies using finer age strata.

As hypothesized, master runners reported significantly higher rates of running participation limitation over the previous seven days as compared to younger runners. Although injury location did not differ substantially between groups for lower extremity joints, master runners were more likely to experience upper body pain, specifically to their shoulders, arms, and forearms. Additionally, significant differences were observed in several demographic, social, and training-related characteristics between age groups. Master runners generally reported higher weekly mileage, longer run training sessions, and greater experience in the sport, alongside distinct equipment habits such as more frequent shoe replacement and use of insoles. While most training characteristics were similar across groups, surface preferences varied, with young runners more likely to train on gravel and master runners more likely to report use of other, unspecified surfaces. Finally, the factors associated with running participation limitation differed by age group, supporting the hypothesis that the determinants of injury-related participation limitation are not uniform across the lifespan. Together, these findings underscore the importance of age-specific considerations when addressing injury prevention and training optimization strategies in the running population.

### Differences in participation limitation between young and master runners

A significantly higher percentage of master runners in the current study reported having running participation limitations over the last seven days as compared to their younger counterparts. This finding aligns with one of the only previous studies comparing injury rates across similar age groups, which found master runners reported an injury rate of 49% over the previous year that affected their ability to train or race, which was significantly higher than younger runners at 45% [19]. These findings suggest that higher rates of RRIs may be associated with reduced ability to maintain consistent running routines, as reflected in the greater participation limitations observed in the current study. This increase in participation limitations among master runners is concerning, particularly against the backdrop of the rising popularity of running. The documented link between running and heightened injury prevalence reflects a broader trend that necessitates targeted prevention strategies for aging runners. It is crucial that these findings inform future approaches to injury risk management, especially considering the differences in running habits and training characteristics in master runners. From a public health perspective, reducing RRIs and the resulting participation restriction may help sustain lifelong physical activity and potentially decrease downstream healthcare utilization associated with recurrent musculoskeletal pain in runners, particularly in the growing master-runner population.

### Differences in location of pain between young and master runners

The injury patterns observed in this study align with prior research, with both young and master runners commonly reporting knee and foot/ankle injuries, reinforcing their status as the most frequently injured body regions across age groups [5,19]. Interestingly, master runners in our study reported a higher proportion of upper extremity injuries. While this finding was statistically significant, the low frequency of such injuries (only 13% of the total cohort) warrants cautious

interpretation, as these cases may not reflect running-specific injuries but rather pre-existing conditions or non-running incidents that cause discomfort during running. Notably, upper extremity pain in older adults may be confounded by higher rates of comorbid musculoskeletal conditions (e.g., shoulder osteoarthritis or rotator cuff tendinopathy) and by non-running activities that commonly load the shoulder/arm (e.g., occupational tasks, resistance training, cycling, falls), which have been shown to contribute substantially to upper-limb symptoms independent of running exposure [35–38]. Still, this observation highlights the need for further investigation into age-related biomechanical changes or training adaptations that could contribute to atypical injury sites and may inform more comprehensive approaches to injury prevention and rehabilitation in master runners.

## Differences in demographics and social habits between young and master runners

The results of this study revealed significant differences between master and younger runners regarding racial identity, educational attainment, and e-cigarette use. The disparities in racial identity highlight potential predispositions to injury that may be influenced by cultural, social, or economic factors within different ethnic groups [39–40]. Notably, the higher education levels observed in master runners could be partially attributed to their older age, which may provide opportunities for advanced education. This may also reflect different access to resources or information regarding injury prevention. Conversely, the increased prevalence of e-cigarette use among younger runners introduces a new potential risk factor that warrants further investigation, as prior work has linked vaping with adverse respiratory outcomes, which could plausibly influence exercise tolerance and performance [41–42]. These findings emphasize the importance of considering demographic and social factors when evaluating injury risk and health behaviors among runners across different age groups.

## Differences in running history and training characteristics between young and master runners

Despite their extensive running histories and commitment to longer training sessions, a higher percentage of master runners reported limitations in their participation due to RRIs, raising important questions about the long-term effects of sustained training on physical health. While years of consistent training enhance cardiovascular fitness and muscular endurance, the repetitive strain of increased mileage and intensity can lead to overuse injuries, as age-related degeneration, such as decreased muscle elasticity and reduced proprioception, heightens susceptibility to conditions like patellar tendinopathy and stress fractures [14,43]. Additionally, the relationship between longer training sessions and fatigue compounds injury risk, as fatigue can impair biomechanical function and lead to altered running mechanics, further exacerbating the likelihood of new injuries [44–45]. Moreover, the cumulative training load experienced over time can lead to a threshold known as the "accumulated fatigue effect," wherein the body has not fully recovered from prior training loads, making a runner more vulnerable to injury during subsequent runs [46–47].

While the practice of owning and frequently replacing multiple pairs of running shoes reflects master runners' awareness of footwear's role in injury prevention [48] and may be related to their financial stability, it may also signify a direct response to these injuries. By rotating footwear and prioritizing mileage over wear and tear, master runners can reduce injury risks associated with overuse and maintain shoe performance [49]. Their use of cushioned insoles might further indicate adaptations made in response to past injuries or healthcare advice. Their continued participation in mid-distance races illustrates their dedication to the sport, yet cannot overshadow the limitations posed by injuries. Collectively, these behaviors, paired with their longer training histories, higher weekly mileage, and more frequent participation in 10 km to half-marathon races, reflect both a greater training load and a proactive approach to injury mitigation in the face of age-related challenges. Given the higher weekly mileage and longer training sessions reported by master runners, clinicians and coaches may consider emphasizing periodized training plans that incorporate planned recovery weeks and symptom-guided load adjustments to better manage cumulative fatigue and reduce participation limitation.

## Differences in training habits between young and master runners

Despite differences in age, training habits between young and master runners were largely similar, particularly regarding warm-up, cool-down, stretching routines, and use of common training surfaces such as asphalt, off-road tracks, and treadmills. However, surface preference differed in two notable ways. Young runners were more likely to train on gravel or pebbles, while master runners more frequently reported training on "other" surfaces. Gravel and pebble surfaces, known for their enhanced shock absorption and reduced impact on joints, can serve as a protective measure for younger runners against some overuse injuries. Research has shown that softer surfaces, like those composed of gravel, can diminish the repetitive stress on the knees and ankles, helping to prevent injuries such as patellar tendinopathy and stress fractures [12]. However, the instability and irregularity inherent in these surfaces can elevate the risk of acute injuries, such as falls and ankle sprains, necessitating careful attention to running technique and form. On the other hand, the preference to run on "other surfaces" observed in master runners may be due to individualized adaptations to minimize impact or accommodate existing injuries. These findings suggest that while foundational training behaviors are consistent across age groups, older runners may make surface-related adjustments to accommodate changing physical needs or injury histories.

## Correlates of running participation limitation in young and master runners

The results of the regression analyses revealed distinct patterns in factors associated with running participation limitation between young and master runners, suggesting age-specific mechanisms underlying injury risk. Among young runners, protective associations with running off-road, having a coach, wearing specialized footwear, and training on multiple surfaces may reflect the benefits of structured training environments, surface variability, and equipment tailored to individual biomechanics, factors that can mitigate overuse and repetitive strain injuries. Conversely, increased risk associated with longer training sessions and running on grass could indicate that exceeding individual training thresholds and running on uneven or softer terrain may predispose younger runners to overload or muscle injuries. However, evidence on grass surfaces is mixed; some studies suggest softer surfaces may reduce impact-related loading, whereas others note that uneven terrain and variability in maintenance may increase acute or soft-tissue injury risk [50–51].

For master runners, the absence of clear protective factors and the identification of risk associations with owning more running shoes, training on gravel or pebbles, and smoking may suggest a more complex injury profile. Rather than prevention, these behaviors could reflect post-injury adaptations or attempts to self-manage chronic injuries. In particular, the association with owning more running shoes may represent reverse causality, whereby runners experiencing recurrent or persistent symptoms purchase/rotate shoes in response to prior injuries rather than shoe ownership contributing to participation limitation. Future studies should explicitly measure and adjust for prior injury history to better disentangle these relationships. Furthermore, using multiple pairs of shoes and choosing specific surfaces may be reactive strategies to manage discomfort, while smoking has been associated with poorer musculoskeletal outcomes and may plausibly impair recovery through mechanisms such as reduced tissue perfusion/oxygenation, altered inflammatory responses, and impaired collagen synthesis, which can delay tissue repair [52–53]. However, this smoking association should be interpreted cautiously because the number of smokers in the master cohort was small (n = 12) and the between-group difference in smoking prevalence did not reach conventional statistical significance (p = 0.06), which may limit estimate stability and increase the possibility of a chance finding. Together, these findings point to the possibility that master runners experience more persistent or systemic barriers to injury recovery and participation, requiring targeted interventions that extend beyond surface-level training modifications.

## Psychosocial considerations and multimodal intervention implications

Psychosocial factors not captured in this survey, particularly age-related differences in motivation (e.g., health/life meaning/affiliation vs. personal goal achievement), as well as traits such as perfectionistic concerns or obsessive passion, may

also influence how master runners respond to symptoms and make participation decisions [18,54]. Related psychological factors (e.g., fear of movement, pain catastrophizing, and coping strategies) can shape pain and functional outcomes, and mindfulness-based approaches have shown promise as an adjunct to exercise therapy for reducing pain and improving function in runners [55]. Recent trials also suggest that multimodal approaches may improve pain-related outcomes in recreational runners, including combining exercise therapy with mindfulness and using multi-component programs (e.g., strengthening, flexibility, neuromuscular control) to reduce overall and overuse RRI incidence [56–58]. Accordingly, clinicians may consider pairing load-modified exercise rehabilitation with adjunct strategies (e.g., mindfulness training and/or targeted strengthening–neuromuscular programs, and condition-specific tools such as foot orthoses when indicated) to support symptom management and return-to-running, while recognizing that these recommendations are informed by emerging interventional evidence and should be tailored to individual presentation and goals.

## Limitations

This study's limitations include several factors that may impact the robustness and generalizability of the findings. First, the reliance on self-reported data from a sample of 207 adult runners could introduce potential recall bias, as participants may misremember or underreport injury prevalence and training habits due to social desirability or lack of awareness. The study design, being cross-sectional, restricts the ability to draw causal inferences regarding the relationship between training behaviors and participation limitations. Additionally, recruitment via social media, university campuses, emails to running clubs, and word-of-mouth may have introduced selection bias by preferentially enrolling runners who are more engaged with running communities and online platforms and/or more motivated to respond due to current symptoms or injury concerns. As a result, injury prevalence and participation limitations may be overestimated and findings may not generalize to less-connected or less-engaged recreational runners. A formal sample size calculation was not performed prior to study initiation. Instead, a minimum of 10 participants per predictor variable was targeted, following established guidelines for logistic regression analyses. With a final sample size of 207 runners, our study was able to include up to 20 predictor variables in the analysis of contributors to participation limitation. Based on the observed strength of associations in the results, the study is adequately powered to support the conclusions drawn regarding predictors of participation limitation in this population. This approach maximized feasibility within available resources while ensuring sufficient analytic capability for the main study objectives.

Also, injury location was self-reported using a body pain diagram. While this approach improves clarity of symptom localization, self-reported pain maps have limited sensitivity and specificity and may not accurately distinguish between soft-tissue, joint, or neurogenic sources of symptoms, potentially leading to misclassification of injury location/type. Additionally, the map did not capture pain intensity or quality. Future studies should consider incorporating clinician assessment and/or imaging (where appropriate) to improve diagnostic accuracy and better characterize injury pathology. Lastly, the study did not extensively explore the psychosocial factors influencing participation limitations, such as mental health, social support, and motivation, which could significantly contribute to understanding injury risk and running behaviors. This suggests a need for further research that integrates these dimensions to better inform injury prevention strategies among expanded runner demographics.

## Conclusion

Understanding the rates and risk factors associated with running participation limitations is crucial for informing runners, coaches, and healthcare professionals about enhancing training practices while minimizing injury risks. As the popularity of running increases, it becomes essential to address injury rates through education and improved training protocols. This study offers insights, particularly regarding aging runners, and underscores the need for future research to adopt a multidimensional approach that examines both physiological and psychological aspects of running. Emphasizing education, tailored training practices, and the unique needs of master runners will help minimize the adverse effects of injury.

Encouraging runners to critically evaluate their training environments, especially concerning surface selection and shoe maintenance, can further mitigate injury risks. Successfully integrating these personalized elements into training strategies will be essential for fostering a sustainable and thriving running culture across all age groups.

## Acknowledgments

We are grateful to all the individuals who participated in this study, and we acknowledge the valuable contribution of their time and personal experiences.

**AI-Assisted Writing Acknowledgment:** Portions of the manuscript were edited for clarity and readability with assistance from ChatGPT (OpenAI). The authors reviewed and revised all AI-assisted text and take full responsibility for the content, accuracy, and interpretation of the work.

## Author contributions

**Data curation:** Jo Armour Smith, Natalia Sánchez.

**Formal analysis:** Andrew Hooyman.

**Investigation:** Jo Armour Smith, Natalia Sánchez, Samantha Jeffcoat, Susan Sigward.

**Project administration:** Shawn Farrokhi.

**Resources:** Shawn Farrokhi.

**Supervision:** Shawn Farrokhi.

**Writing – original draft:** Rachel Berns.

**Writing – review & editing:** Rachel Berns, Jo Armour Smith, Natalia Sánchez, Samantha Jeffcoat, Susan Sigward, Andrew Hooyman, Shawn Farrokhi.

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
