## [Decision Letter · Decision Letter 0]

27 Jan 2026

PONE-D-25-60207Variation in prevalence of participation limitation, injury location, and training habits between young and master runners: a cross-sectional studyPLOS One

Dear Dr. Berns,

Thank you for submitting your manuscript to PLOS ONE. After careful consideration, we feel that it has merit but does not fully meet PLOS ONE’s publication criteria as it currently stands. Therefore, we invite you to submit a revised version of the manuscript that addresses the points raised during the review process.

We look forward to receiving your revised manuscript.

Kind regards,

Aynollah Naderi

Academic Editor

PLOS One

Journal Requirements:

2. Please note that your Data Availability Statement is currently missing the repository name. If your manuscript is accepted for publication, you will be asked to provide these details on a very short timeline. We therefore suggest that you provide this information now, though we will not hold up the peer review process if you are unable.

Reviewers' comments:

Reviewer's Responses to Questions

**Comments to the Author**

1. Is the manuscript technically sound, and do the data support the conclusions?

Reviewer #1: Yes

Reviewer #2: Yes

2. Has the statistical analysis been performed appropriately and rigorously? 

Reviewer #1: Yes

Reviewer #2: Yes

3. Have the authors made all data underlying the findings in their manuscript fully available?

Reviewer #1: Yes

Reviewer #2: Yes

4. Is the manuscript presented in an intelligible fashion and written in standard English?

Reviewer #1: Yes

Reviewer #2: Yes

5. Review Comments to the Author

Reviewer #1: Background provided sufficient rationale for need for this research; purpose statement and hypotheses were clear. Methods was comprehensive and well organized. Results, discussion, and conclusion were consistent with the study's purpose; some brief additional statements are recommended for the initial discussion. Only very minor revisions are suggested. See pdf for specific comments.

Reviewer #2: Thank you for submitting your manuscript entitled “Variation in prevalence of participation limitation, injury location, and training habits between young and master runners: a cross-sectional study” to PLOS ONE. Your cross-sectional investigation, based on survey data from 207 runners and employing the Oslo Sports Trauma Research Center Overuse Injury Questionnaire, provides valuable comparative insights into age-related differences in running-related participation limitations and associated factors. The findings regarding higher limitations in master runners (>35 years), increased upper extremity injuries, and distinct risk profiles contribute meaningfully to the literature on recreational running and support the call for age-specific prevention strategies.

The study has several strengths, including the use of a validated overuse injury tool, transparent reporting of ethics and data availability (via OSF), and clear presentation of age-specific LASSO regression results. However, several methodological and interpretive issues require attention to enhance scientific rigor and generalizability.

6. PLOS authors have the option to publish the peer review history of their article (what does this mean?). If published, this will include your full peer review and any attached files.

Reviewer #1: No

Reviewer #2: No

---

## [Author Response · Author response to Decision Letter 1]

11 Mar 2026

All responses to specific reviewer/editor comments have been provided in the document titled "Response to Reviewers.docx". Thank you for your time and consideration!

---

## [Editor Report · Decision Letter 1]

27 Mar 2026

Variation in prevalence of participation limitation, injury location, and training habits between young and master runners: a cross-sectional study

PONE-D-25-60207R1

Dear Dr. Rachel Berns,

We’re pleased to inform you that your manuscript has been judged scientifically suitable for publication and will be formally accepted for publication once it meets all outstanding technical requirements.

Kind regards,

Aynollah Naderi

Academic Editor

PLOS One

Additional Editor Comments (optional):

Dear Authors,

Thank you for submitting your revised manuscript to PLOS ONE. I have reviewed your responses to the reviewers' comments and the updated manuscript.

I am pleased to inform you that your manuscript is recommended for acceptance in its current form.

Your study provides valuable, age-stratified insights into running-related participation limitations and injury patterns. The use of the validated OSTRC-O2 questionnaire, transparent reporting, and appropriate application of LASSO regression strengthen the methodological rigor. The authors have comprehensively addressed all reviewer concerns, clarifying key methodological choices (e.g., OSTRC-O2 dichotomization, LASSO justification) and enhancing the discussion with practical, age-specific implications for injury prevention.

Minor production notes: please ensure table footnotes are consistent and figure labels are optimized for accessibility.

Congratulations on this contribution to sports medicine literature. Thank you for choosing PLOS ONE.

Sincerely,

Aynollah Naderi, PhD

Associate Professor, Sport Sciences

Shahrood University of Technology, Iran
---

## [Editor Report · Acceptance letter]

PONE-D-25-60207R1

PLOS One

Dear Dr. Berns,

I'm pleased to inform you that your manuscript has been deemed suitable for publication in PLOS One. Congratulations! Your manuscript is now being handed over to our production team.

Kind regards,

on behalf of

Dr. Aynollah Naderi

Academic Editor

PLOS One